# Hour of Life at Enteral Feeding Initiation and Associated Clinical Morbidity in Extremely Low-Birth-Weight Infants

**DOI:** 10.3390/nu16234041

**Published:** 2024-11-26

**Authors:** Melissa Thoene, Lauren Ridgway, Elizabeth Lyden, Ann Anderson-Berry

**Affiliations:** 1Department of Pediatrics, Division of Neonatology, University of Nebraska Medical Center, 981205 Nebraska Medical Center, Omaha, NE 68198, USA; 2College of Public Health, University of Nebraska Medical Center, Omaha, NE 68198, USA

**Keywords:** enteral feeding, extremely low birth weight

## Abstract

Background/Objectives: Identifying nutritional interventions in extremely low-birth-weight (ELBW) infants (<1000 g) that are associated with favorable clinical outcomes is important. Delayed enteral feeding initiation (>3 days) has been associated with increased odds of developing morbidity. Therefore, the aim of this study is to evaluate the relationship between hour of life at enteral feeding initiation and associated clinical outcomes. Methods: An IRB-approved retrospective chart review evaluated ELBW infants. Birth acuity was evaluated using CRIB II scoring and incidence of various morbidities (bronchopulmonary dysplasia (BPD), retinopathy of prematurity (ROP), necrotizing enterocolitis (NEC), and spontaneous intestinal perforation (SIP)) and mortality was assessed after adjustment. *p* < 0.05 was statistically significant. Results: A total of 27/61 (44.3%) initiated enteral feeding <12 h of life. CRIB II scores were lower in infants with earlier enteral feeding initiation. There were no statistical differences in NEC, SIP, or death between categories of hour of life at enteral feeding initiation. After adjusting for CRIB II scores, enteral feeding initiation ≥12 h of life was associated with more days receiving oxygen >21% inspired air (β = 32.7; *p* = 0.040), approximately 7-fold higher odds of developing moderate/severe BPD (95% CI 1.2.8–38.28; *p* = 0.025), and 9-fold higher odds of being discharged home while receiving oxygen therapy (95% CI 1.03–79.81; *p* = 0.047). Conclusions: Timing of enteral feeding initiation may be delayed in ELBW infants with higher clinical acuity, yet later initiation by hour of life is associated with worsened clinical respiratory outcomes. Early initiation within the first 12 h of life is feasible and was not associated with gastrointestinal morbidity in this single-center cohort of ELBW infants.

## 1. Introduction

Extremely low-birth-weight (ELBW) infants (<1000 g) are inherently at risk for higher incidence of morbidity and mortality due to their extremely small size and degree of prematurity. Many variables contribute to the risk for developing morbidities like bronchopulmonary dysplasia (BPD) or retinopathy of prematurity (ROP). Many of these variables are non-modifiable after preterm birth including infant sex, weight plotting of small for gestational age, or maternal prenatal variables like pre-delivery steroid administration, hypertension, or infection [1]. Yet, these multifactorial diseases are also associated with modifiable interventions, therefore allowing opportunity for enhanced care protocols which may ultimately lessen disease progression.

ELBW infants have highly specialized nutritional needs to support proper growth and development. Meeting these specialized needs is critical to promote enhanced neurodevelopmental outcomes and to lower risk or severity of morbidities associated with preterm birth [2,3]. In historical observation of nutrition provision, increased energy provision within the first one month of life has been associated with higher developmental index scoring and decreases in BPD by 9% and severe ROP by 24%, even with small incremental increases by an average of 10 calories (kcal)/kilogram (kg)/day [4,5,6]. This research clearly demonstrates how small yet feasible nutritional alterations may have substantial impact on long-term outcomes in this high-risk population. Thus, there is significant opportunity for early and favorable impact through clinical care by focusing on the minute practices of nutrition support management. In addition to evaluating global provision of macro- and micronutrient intake, timing and route of delivery are also important variables to consider.

Though incompletely developed, the fetal gastrointestinal tract is in use prior to birth with the swallowing of amniotic fluid [7]. Markedly, as amniotic fluid contains essential nutrients, this swallowing of amniotic fluid allows for intestinal absorption and may contribute up to 10–15% in meeting fetal nutritional needs [8,9,10]. Thus, provision of enteral substrate after preterm birth is beneficial to promote physical, physiological, and microbiome development. While the benefits of enteral feeding in preterm infants are numerous [11], even the use of human milk for oral care has been associated with clinical benefit, including a decreased risk of ventilator-associated pneumonia and necrotizing enterocolitis (NEC) and a shortened time for receiving mechanical ventilation [12,13]. Though it is beneficial to provide enteral substrate, these data still suggest that the comprehensive benefits of early enteral feeding are not yet sufficiently elucidated.

Timing of providing enteral substrate may also impact infant health and clinical outcomes. However, high clinical acuity, clinician fear of adverse outcomes, and varying unit practices may delay introduction in ELBW infants. Data from term newborns showed that initiation of breastfeeding >1 h after birth is associated with worsened health in the first 6 months of life, including upper respiratory infection symptoms and cough [14]. In a preterm population, Konnikova et al. reported delayed initiation of enteral feeding (defined as initiating enteral feeding >3 days of age) in infants born <33 weeks gestation was associated with a 4.5-fold increase in chronic lung disease (95% CI 1.8–11.5, *p* = 0.002), 2.9-fold increase in ROP (1.1–7.8, *p* = 0.03), and 3.4-fold increase in multiple comorbidities (1.2–9.8, *p* = 0.02) as compared to early initiation of enteral feeding (at <3 days of age) [15]. Likewise, it was reported that delayed initiation of enteral feeding was associated with increased fecal markers of inflammation [15]. As intestinal villous atrophy may incur following even a brief period of no enteral feeding, the evaluation of initiating enteral feeding within days after birth may span too large a gap when considering the association with clinical outcomes. Thus, current knowledge is limited on how hour of life at enteral feeding initiation in preterm infants is associated with later health and clinical outcomes. Therefore, the purpose of this study is to evaluate the relationship between the hour of life at which enteral feeding is initiated and associated comorbidities in a population of ELBW infants.

## 2. Materials and Methods

The local Institutional Review Board approved this study (#0665-19-EP; originally approved 3 December 2019 with last reapproval 28 March 2024) with a waiver of parental consent, given the data were evaluated retrospectively. Therefore, included subjects were all ELBW infants (<1000 g, *n* = 61) born in a level III NICU at the University of Nebraska Medical Center (Omaha, NE, USA), regardless of gestational age between 1 June 2016 to 15 April 2020. Excluded infants were those who were born with major congenital anomalies or transferred to another hospital within the first 30 days of life. There was no exclusion based on any maternal factors. Data were collected from electronic medical records of included subjects between 3 December 2020 to 20 October 2022. The authors had access to identifiable individual records during data collection to ensure accuracy, but collected data were de-identified prior to statistical analysis and interpretation.

A retrospective chart review was conducted using the electronic medical record (EMR). Acuity at birth was evaluated by calculating Clinical Risk Index for Babies (CRIB) II scores [16,17]. Incidence of mortality and neonatal morbidities were assessed including spontaneous intestinal perforation (SIP), BPD as defined by the National Institutes of Child Health and Disease (NICHD) criteria at 36 weeks gestation [18], necrotizing enterocolitis (NEC) as defined and categorized using modified Bell’s staging criteria [19], and ROP with information obtained from ophthalmologist evaluation using the revised international classification for ROP [20]. Additional demographic and clinical data during hospitalization in the neonatal intensive care unit were collected including race and ethnicity, birth gestational age, birth weight, number of days receiving mechanical ventilation, and number of days receiving oxygen support at >21% inspired oxygen. Growth data were obtained and compared using the 2013 Fenton preterm infant growth chart [21].

The hour of life at which enteral feeding was initiated and substrate type were determined based on medical documentation of first feeding delivery. The hour of life at which enteral feeding was initiated was evaluated as a continuous variable but was also categorized as initiation at <12 h, 12–24 h, and >24 h of life. Type of substrate (mother’s own milk, pasteurized donor human milk, or preterm infant formula) was evaluated at varying time points including at enteral feeding initiation and discharge. Full-volume enteral feeding was defined as the infant receiving at least 145 mL/kg/day of enteral nutrition (and no parenteral nutrition), with human milk fortified to 24 kcal/oz using a commercial human milk fortifier and the addition of a protein modular, with details included in the following section.

### 2.1. Nutrition Support Management

Nutrition management practices are standardized in the unit for which these infants were cared for; however, practices may vary based on the discretion of the medical team providing individualized care.

ELBW infants in this cohort were initiated on a starting standard parenteral nutrition solution (10% dextrose, 4% amino acids) immediately after birth once intravenous lines (via an umbilical venous catheter) were placed. If an umbilical arterial catheter was simultaneously place, a solution exclusively containing amino acids (3.2% amino acids) was infused at 0.5 mL/h continuously to keep the line patent. The provided parenteral volume of these solutions was initiated at a goal volume of 80 milliliters (mL)/kg/day on birth weight with a glucose infusion rate between 4 and 6 milligrams (mg)/kg/minute but was subsequently adjusted if indicated based on hydration or blood glucose laboratory values. Custom parenteral nutrition was initiated as soon as feasible within the first 24–36 h of life. Intralipids were initiated at 1.5–2.0 g/kg/day on the first day of custom parenteral nutrition, then increased over the next 1–2 days to a goal of 3.0 g/kg/day as individual clinical lab results (e.g., triglyceride levels) allowed. Protein was provided at the start of custom parenteral nutrition at a goal of 4.0 g/kg/day on birth weight. The glucose infusion rate was increased daily as blood glucose values allowed. The minimum goal parenteral energy target was 100 kcal/kg/day (based on birth weight during the first week of life), though total daily energy goals were increased in combination with the dual provision of enteral feeding.

Enteral nutrition was initiated via nasogastric or orogastric tube as soon as medically appropriate after birth, with the ideal goal being within the first 4 h of life, as per unit protocol, to allow opportunity for the first feeding to use the mother’s own milk. However, the start of enteral feeding is based on neonatologist discretion, though unit contraindications included dopamine or pressors >10 mcg/kg/minute, epinephrine or norepinephrine >0.1 mcg/kg/minute, hemodynamic instability, inability to appropriately oxygenate, and pH persistently less than 7.0. Enteral feeding was initiated at a targeted 30–35 mL/kg/day [22] based on birth weight as continuous infusion [23]. Trophic feeding at 30–35 mL/kg/day was standardly continued for 48 h in infants born <28 weeks gestation and only 24 h in infants born >28 weeks gestation. Subsequently, enteral feeding volume was increased daily by 30–35 mL/kg/day to a preferential unit goal of 150 mL/kg/day on birth weight. Human milk fortification using a liquid, non-acidified, bovine protein-based human milk fortifier [24,25] was initiated when enteral volumes reached 50–60 mL/kg/day on birth weight [26]. Enteral feeding fortification and volume increases occurred simultaneously. Infants <26 weeks gestation, initiated human milk fortification at 22 kcal/oz for 1–2 days before advancing to 24 kcal/oz. After tolerating 24 kcal/oz for at least 24 h, a hydrolyzed liquid protein modular (containing 1 g of protein per 6 mL) [27] was added to achieve goal targets of 120 kcal/kg/day and a minimum of 4 g protein/kg/day at an enteral volume of 150 mL/kg/day. If growth was inadequate while on 24 kcal/oz fortified feedings, human milk fortification was increased to 27 or 30 kcal/oz by the addition of a preterm infant formula powder [28] and the addition of liquid protein adjusted as needed to meet the goal minimum of 4 g protein/kg/day.

Pasteurized donor human milk obtained from an established human milk bank following Human Milk Banking Association of North America (HMBANA) guidelines was available for use during the first 14 days of life if the parent/s provided assent and the mother’s own milk was unavailable, of limited quantity, or contraindicated for use. The donor human milk was fortified equivocally as the mother’s own milk. If inadequate volume of MBM was available by day of life 14, infants were transitioned over 2 days from pasteurized donor human milk to supplementation with a preterm infant formula (24 kcal/oz ready-to-feed containing 3.6 g protein per 100 kcal) [29]. For parents not providing assent for use of donor human milk, enteral feeding was provided as this same preterm formula if insufficient mother’s milk was available. The caloric density of the provided preterm formula was increased to 27 or 30 kcal/oz [30] if needed to promote appropriate growth. Targeted goal weight gain to maintain growth percentiles and z-scores during this study period was commonly ~18–21 g/kg/day for infants <2 kg [3] and ~30–35 g/day once >2 kg until term age (minimum 37 weeks gestation), then transitioned to 25–35 g/day [31]. Weight gain from birth to hospital discharge was calculated in grams/day and grams/kg/day using the exponential formula by Patel et al. [32].

All infants received sublingual concentrated vitamin D3 supplementation of 800 international units per day during hospitalization from day of birth until discharge or term-corrected age. In addition, a probiotic supplement (0.5 g/day; 1 million colony-forming units) containing five bacterial strains (*Bifidobacterium breve*, *Bifidobacterium longum*, *Bifidobacterium infantis*, *Bifidobacterium bifidum*, and *Lactobacillus rhamnosus*) was administered once enteral feeding volume reached minimum 48 mL/day. Daily enteral iron fortification (ferrous sulfate, dosing range 2–4 mg/kg/day) was initiated at 14 days of life primarily for infants consistently receiving at least half of their enteral feeding volume as their mother’s own milk.

### 2.2. Statistical Analysis

Descriptive statistics include medians, interquartile ranges (IQRs), and minimums and maximums for each continuous data sample and counts and percentages for categorical data samples. Fisher’s exact test compared categorical clinical characteristics and outcomes between groups with different times of enteral feeding initiation. The Kruskal–Wallis test compared outcomes that were continuous measures between the three groups of enteral feeding timing. Spearman correlation coefficients assessed the relationships between hour of life at which enteral feeding was initiated and continuous birth and outcome variables.

Multivariable linear regression with either hour of life at enteral feeding initiation as a continuous measure or dichotomized at 12 h and CRIB II score were used as predictors of continuous outcomes. Multivariable logistic regression either with hour of life at enteral feeding initiation as a continuous measure or dichotomized at 12 h and CRIB II score were used as predictors of dichotomous outcomes. The Firth adjustment was used in conjunction with the multivariable logistic regression models to stabilize the odds ratio estimates and confidence intervals because the outcome rates were small. All analyses were completed using SAS 9.4. A *p*-value < 0.05 was considered statistically significant for all tests.

## 3. Results

This population of ELBW infants (*n* = 61) was born between 22 and 29 weeks gestation. Demographic and clinical outcome data are reported in Table 1 and Table 2. A total of 83.6% of all infants initiated enteral feeding within the first 24 h of life and 44.3% initiated at <12 h after birth. Furthermore, 44 infants (72.1%) were able to achieve full enteral feeding. Among all infants, 32.8% (*n* = 20) were able to have the first documented feeding as their mother’s own milk, with most of the remainder initiating feeding with donor human milk (63.9%), followed by formula (3.2%). At 28 days of life, for those receiving enteral feeding, 55.0% (*n* = 22/40) were receiving any quantity of their mother’s own milk. Similarly, 57.9% (*n* = 22/38) were receiving any quantity of their mother’s own milk at 36 0/7 weeks gestation, which decreased to 37.8% (*n* = 14/37) by time of hospital discharge.

Infants were categorized by the hour of life at which enteral feeding was initiated to compare demographic and clinical outcomes (Table 3). There was no significant difference in NEC, SIP, or death between categories of enteral feeding initiation. CRIB II scores were lower in infants with the earliest initiation of enteral feeding (<12 h), and they clinically achieved full enteral feeding more quickly with results approaching significance (*p* = 0.0971). Likewise, infants initiating enteral feedings the earliest had lower incidence of ROP, BPD, and fewer days receiving oxygen support at >21% inspired air and days of mechanical ventilation. There were no significant differences observed with timing to achieve full enteral feeding, infant sex, intrauterine growth restriction, or any maternal factors including race, receipt of steroids before delivery, smoking, diabetes during pregnancy, or having any form of high blood pressure during pregnancy. All three categories for timing of enteral feeding initiation had no difference in receiving their mother’s own milk at time of reaching full enteral feeding, at 28 days of age, at 36 0/7 weeks gestational age, or at discharge. However, there was a trend across groups for significance in the percentage of infants receiving their mother’s own milk as the first documented feeding (<12 h = 7/27 (25.9%), 12–24 h = 6/24 (25.0%), and >24 h 7/10 (70.0%); *p* = 0.0857).

When evaluating hour of life at enteral feeding initiation (as a continuous variable) among surviving infants and after adjusting for CRIB II scores, there was no association with developing any form of ROP (*p* = 0.639) or BPD (*p* = 0.638), or developing ROP > Stage 2 (*p* = 0.342) or moderate or severe BPD (*p* = 0.859). Furthermore, it did not have an association with being on oxygen support at 28 days of age (*p* = 0.638) or being discharged home on oxygen (*p* = 0.851).

Similarly, when evaluating hour of life at enteral feeding initiation among surviving infants after adjusting for CRIB II scores, we observed no significance with days receiving oxygen at >21% inspired air (*p* = 0.455) or days receiving mechanical ventilation (*p* = 0.875). Further, there was no observed significance with day of life to achieve full enteral feeding (*p* = 0.530), discharge gestational age (*p* = 0.391), or hospital length of stay in days (*p* = 0.570). Hour of life at enteral feeding initiation was associated with birth to discharge z-score change for head circumference (β= −0.025; *p* = 0.0012) and weight (β= −0.008; *p* = 0.038) but not length (*p* = 0.572).

Lastly, the relationship between dichotomized timing of enteral feeding initiation (≥12 h vs. <12 h) and clinical morbidity was further evaluated in surviving infants after adjusting for CRIB II scores (Table 4). The results showed no difference in the odds of developing any stage or severity of ROP or BPD, or developing ROP at >Stage 2. However, enteral feeding initiation at ≥12 h was associated with higher odds of developing moderate/severe BPD (OR 6.99, 95% CI 1.2.8–38.28; *p* = 0.025) and being discharged home while receiving oxygen therapy (OR 9.08, 95% CI 1.03–79.81; *p* = 0.047). Likewise, initiating enteral feeding at ≥12 h was associated with increased number of days receiving oxygen at >21% inspired air (β = 32.7; *p* = 0.040) and approached significance with discharge CGA (β = 2.89; *p* = 0.094). Initiating enteral feeding at ≥12 h also approached significance with birth to discharge length z-score change (β= −0.65; *p* = 0.099) but had no effect on timing to reach full enteral feeding (*p* = 0.240), days receiving mechanical ventilation (*p* = 0.708), hospital length of stay (*p* = 0.106), or birth to discharge z-score change for head circumference (*p* = 0.737) or weight (*p* = 0.805).

## 4. Discussion

In this cohort of ELBW infants, the results showed that early initiation of enteral feeding is feasible—even at <12 h after birth—but timing may be influenced by clinical acuity. This finding is not unexpected in a population of ELBW infants who may experience respiratory failure or hemodynamic instability after birth. Yet, after adjustment for CRIB II scores (our selected surrogate indicator for clinical acuity after birth), results suggest that earlier initiation of enteral feeding (even by hours) may have significant and potentially long-term influence on the outcomes of this high-risk population. Our results mimic those by Konnikova et al. who likewise reported increased odds for morbidities like chronic lung disease in infants born <33 weeks gestation who initiated enteral feedings >3 days of age [15]. Yet, the novelty of our results more minutely elucidates the time-sensitive association between clinical outcomes and enteral feeding initiation, which is a modifiable intervention.

Overall, this cohort of ELBW infants had high incidence of mortality at 39.3% (75% primarily attributed to cardiopulmonary failure) and collective morbidity including 84.2% BDP (*n* = 32/38), 71.8% ROP (*n* = 28/39), 8.2% NEC, and 4.9% SIP. The incidence of mortality was higher than average for our unit during this period of evaluation, with this subset including a few unique and complex patient cases, even resulting in one late death at 271 days of life. Beyond mortality, there was no significant difference in NEC or SIP between categories of hour of life at enteral feeding initiation. However, clinically, incidence was highest in the 12–24 h group compared to <12 or >24 h groups. This may be of random incidence within a smaller-sized cohort or may be attributed to variables not examined within this study. Yet, this lack of significant differences in gastrointestinal-related complications of prematurity between categories for timing of enteral feeding initiation is important to consider. Though additional evaluation in a larger-sized cohort is warranted, this study suggests that delayed initiation of small volume enteral feeding in ELBW infants may not decrease the feared risk of gastrointestinal complications. Further, results must be considered with data reported by a systematic review in ELBW infants which concluded that initiation of enteral feedings in the first 3 days of age is associated with decreased risk of SIP [33]. However, more research is needed on decreasing viable birth sizes and gestational ages of the ELBW infant population, as contrasting literature reports a decreased incidence of NEC and associated surgical intervention or death when delaying enteral feeding initiation until 10–14 days of life in infants born <750 g [34]. Yet, a prior Cochrane review demonstrated no protective effects of delayed enteral feeding initiation (often defined as initiation at >4–7 days of age) in a collective population of very low-birth-weight infants [35], but timing has yet to be fully evaluated in hours compared to days. Likewise, the combined impact of timing of enteral feeding initiation alongside additional considerations like provision of oral care using human milk and volume of enteral feeding (in mL/kg/day) after initiation must be evaluated more completely in relation to morbidity and timing to achieve full oral feeding. Still, given our results demonstrated an association with respiratory outcomes, early initiation of enteral feeding must be evaluated in each unit and considered carefully when defining standard-of-care practices.

Nonetheless, and after adjusting for clinical birth acuity via CRIB II scores, the results still showed significantly increased odds of worsened clinical outcomes for infants initiating enteral feeding at ≥12 h of life. This included approximately 7-fold increased odds of developing moderate or severe BPD and 9-fold increased odds of being discharged home while receiving oxygen therapy. Though additional impactful variables were not evaluated, infants initiating enteral feeding at ≥12 h of life were predicted, on average, to receive an additional 32.7 days of oxygen at >21% inspired air. Though not statistically significant, a clinically relevant consideration is that infants who initiated enteral feeding at ≥12 h of life were discharged, on average, approximately 2.89 weeks later than infants initiating enteral feeding at <12 h. These many outcomes have clinical ramifications and generally contribute to increased costs of medical care, worsened neurodevelopment or physical health, or prolonged NICU hospitalizations. Thus, among neonatal populations inherently at increased risk for developing clinical morbidities, modifiable strategies must be evaluated and employed if able to favorably influence disease progression or severity.

Our results also showed that later initiation of enteral feeding in hours of life (as a continuous variable and after adjusting for CRIB II scores) was significantly associated with larger decreases in infant birth to discharge z-score change for weight and head circumference. However, when evaluating timing of enteral feeding initiation as a dichotomous variable, there were no statistical differences. Yet, it was clinically predicted that length z-score would decline an additional 0.65 for infants initiating enteral feeding at ≥12 compared to <12 h of life (*p* = 0.99). Contributors of these findings are likely multifactorial and likely include both nutrition-related and non-nutrition-related factors. Notable, however, is that these growth findings coincide with a higher observed incidence of respiratory morbidity in infants initiating enteral feeding at ≥12 h of life. Infants medically requiring increased or prolonged respiratory support, as well as those developing moderate or severe BPD, may conjunctively receive interventions or medications that impair normal linear growth such as steroids [36]. Likewise, clinical illness may also impact linear growth and associated fat-free mass, with reductions in both associated with worsened neurodevelopmental outcomes in preterm infants [37,38]. With regard to these multifactorial considerations, our results demonstrating significant relationships between timing of enteral feeding initiation and longitudinal growth outcomes highlight the importance of individual units evaluating outcomes resulting from both singular and comprehensive medical care interventions among the ELBW infant population.

The number of days for infants receiving parenteral nutrition (PN) prior to reaching full enteral feeding was not statistically different across groups based on timing of enteral feeding initiation. However, this study did not evaluate detailed volume, macronutrient, or micronutrient delivery early in life (from PN, enteral nutrition, or a combination of both) between analyzed groups, which have been associated with developing morbidities [4,5,6]. Likewise, less than half of evaluated surviving infants were receiving their mother’s own milk by the time of discharge, which may additionally impact the development of clinical outcomes such as moderate or severe BPD [39]. While this study contributes foundational information on the time-sensitive relationship between enteral feeding initiation and clinical morbidity, future studies among a larger cohort of ELBW infants should evaluate this same relationship after multivariate adjustment for clinical acuity, substrate type, and detailed nutrient intake.

### 4.1. Strengths

The primary strength of this study is its novelty in examining associations with clinical outcomes in ELBW infants and timing of enteral feeding initiation in hours as opposed to days. Significant results, or those approaching significance, are clinically important to consider. Further research is therefore indicated and units caring for ELBW infants are encouraged to critically evaluate current enteral nutrition practices and protocols to identify modifiable opportunities to promote enhanced outcomes.

### 4.2. Limitations

Limitations of this study include a small sample size evaluated within a single center, making results less generalizable to all populations of ELBW infants. Further, reported study results did not account for additional factors beyond CRIB II scores that may impact clinical outcomes and/or risk of developing morbidity such as detailed nutrient intake, provision of oral care with human milk, volume (in mL/kg/day) of enteral feeding after initiation, medication administration, clinical care within the first hour of life, ventilation requirements, infant sex, or detailed maternal pregnancy and delivery information. Furthermore, information collected for this retrospective study relied upon data entered into the electronic medical record by multiple health care providers which was later evaluated and collected by the research team, all of which may incur variations between individuals for actual vs. reported or perceived interpretation. Yet, despite these limitations, this research demonstrates feasible methods for units retrospectively analyzing nutrition management practices, especially in the context of quality improvement initiatives.

## 5. Conclusions

Timing of enteral feeding initiation may be delayed in ELBW infants with higher clinical acuity. However, after adjusting for CRIB II scores, enteral feeding initiation at ≥12 h of life was observed to be associated with more days receiving oxygen at >21% inspired air and approximately 7-fold higher odds of developing moderate or severe BPD and 9-fold higher odds of being discharged home while receiving oxygen therapy. Likewise, early initiation of enteral feeding after birth within the first 12 h of life was feasible in this single-center population of ELBW infants and was not statistically associated with gastrointestinal morbidity. Thus, modifiable decisions regarding enteral feeding initiation after birth are time-sensitive and may importantly impact longitudinal health.

## Figures and Tables

**Table 1 nutrients-16-04041-t001:** Categorical birth demographics and clinical outcome data (*n* = 61).

Variable	Category	*n* (%)
Maternal Race and Ethnicity	Black	1 (1.6%)
	White	15 (24.6%)
	Hispanic	5 (8.2%)
	Other/Unknown	19 (31.1%)
	Missing Data	21 (34.4%)
Maternal Smoking Status	Never a Smoker	45 (73.8%)
	Active/Former Smoker	16 (26.2%)
Maternal Diabetes during Pregnancy	Yes	9 (14.8%)
	No	52 (85.2%)
Maternal High Blood Pressure during Pregnancy (any type)	Yes	30 (49.2%)
	No	31 (50.8%)
Mother Received Steroids Before Delivery	Yes	56 (91.8%)
	No	5 (8.2%)
Infant Born Small for Gestational Age *	Yes	12 (19.7%)
	No	49 (80.3%)
Infant Born Intrauterine Growth-Restricted	Yes	11 (18.0%)
	No	29 (47.5%)
	Missing Data	21 (34.4%)
Infant Sex	Male	37 (60.7%)
	Female	24 (39.3%)
Infant CRIB II Score Category at Birth	Score 6–10	20 (32.8%)
	Score 11–15	33 (54.1%)
	Score > 15	8 (13.1%)
Hour of Life at Enteral Feeding Initiation	<12	27 (44.3%)
	12–24	24 (39.3%)
	>24	10 (16.4%)
Bronchopulmonary Dysplasia at 36 0/7 Weeks (*n* = 38)	No	6 (15.8%)
	Mild	14 (36.8%)
	Moderate	2 (5.3%)
	Severe	16 (42.1%)
On Oxygen Support at 28 Days of Life (*n* = 40)	Yes	34 (85.0%)
	No	6 (15.0%)
Discharged Home on Oxygen Support (*n* = 37)	Yes	12 (32.4%)
	No	25 (67.6%)
Retinopathy of Prematurity (*n* = 39)	Yes	28 (71.8%)
	No	11 (28.2%)
Highest Stage of Retinopathy of Prematurity (*n* = 39)	0	11 (28.2%)
	1	14 (35.9%)
	2	6 (15.4%)
	3	7 (17.9%)
	4	1 (2.6%)
Intraventricular Hemorrhage	Yes	20 (32.8%)
	No	25 (63.9%)
	Missing Data	2 (3.3%)
Necrotizing Enterocolitis	Yes	5 (8.2%)
	No	56 (91.8%)
Spontaneous Intestinal Perforation	Yes	3 (4.9%)
	No	58 (95.1%)
Infant Death	Yes	24 (39.3%)
	No	37 (60.7%)
Cause of Death (*n* = 24)	Cardiopulmonary Failure	18 (75.0%)
	Gastrointestinal Disease	3 (12.5%)
	Infection	2 (8.3%)
	Withdrawal of Care	1 (4.2%)

* Birth weight plotting <10th% (or z-score < −1.28) for age on the 2013 Fenton preterm infant growth chart.

**Table 2 nutrients-16-04041-t002:** Continuous birth demographics and clinical outcome data (*n* = 61).

Variable	*n*	Median (IQR)	Minimum	Maximum
Birth Gestational Age (weeks)	61	25.4 (24.0–27.4)	22.9	30.0
Birth Weight (g)	61	710 (570–850)	400	990
Birth Length (cm) and Z-score	61	31.0 (29.0–33.0)	20.0	40.0
Birth Head Circumference (cm)	61	22.0 (21.0–24.0)	18.6	26.5
Discharge Gestational Age (weeks)	40	42.6 (39.0–46.1)	28.7	62.6
Length of Hospital Stay (days)	40	115.5 (74–149)	35	278
Discharge Weight (g)	40	3712 (2774–4281)	820	5967
Discharge Length (cm)	40	49.1 (46.3–51.0)	32.0	56.5
Discharge Head Circumference (cm)	40	35.5 (33.5–37.0)	23.3	39.0
Birth to Discharge Weight Z-score Change *	40	−0.5 (−0.9 to −0.1)	−3.9	1.0
Birth to Discharge Length Z-score Change *	40	−0.8 (−1.5 to −0.3)	−5.5	1.3
Birth to Discharge Head Circumference Z-score Change *	40	0 (−0.6 to 0.9)	−4.6	2.3
Weight Gain during Hospitalization (g/day)	40	24.4 (22.9–25.7)	9.5	30.8
Weight Gain during Hospitalization (g/kg/day)	40	13.8 (12.8–15.0)	9.6	19.7
CRIB II Score at Birth	61	13.0 (10–14)	6.0	18.0
Hour of Life at Enteral Feeding Initiation	61	12.7 (8.7–19.3)	3.4	178.4
Day of Life to Full Enteral Feeding	44	10 (8–20)	6	107
Total Number of Days Receiving Parenteral Nutrition During Hospitalization	49	9 (7–20)	1	168
Days Receiving Oxygen at >21% Inspired Air	40	59.5 (38–104.5)	0	278
Days Receiving Mechanical Ventilation	40	33.5 (2.5–63.5)	0	278
Day of Life at Death	24	11.5 (6.0–13.5)	1	271

* Z-scores based on the 2013 Fenton preterm infant growth chart. Therefore, for infants discharging at >50 0/7 weeks, “discharge” z-score (and birth to discharge change in z-score) is based on the z-score for the last anthropometric measurement available at ≤50 0/7 weeks gestational age.

**Table 3 nutrients-16-04041-t003:** Timing of enteral feeding initiation and association with birth demographics and clinical outcomes.

		Enteral Feeding Initiation Category			
Variable	*n*	<12 h (*n* = 27)	12–24 h (*n* = 24)	>24 h (*n* = 10)	*p*-Value Unadjusted, Across Groups
Mother Received Steroids before Delivery	61	Yes 26 (96.3%)	Yes 20 (83.3%)	Yes 10 (100%)	0.181
Median Birth Gestational Age (weeks)	61	26.0	25.2	24.8	0.403
Median Birth Weight (g)	61	735	725	610	0.091
Median Discharge Gestational Age	40	40.0	44.4	46.9	0.057
Median Hospital Length of Stay (days)	40	80.0	124.0	149.0	0.092
Median Hour of Life at Enteral Feeding Initiation	61	8.6	16.9	27.6	<0.0001
Median Day of Life to Full Enteral Feeding	44	8.5	12.0	19.0	0.112
CRIB II Score Category	61				
Score 6–10		11/27 (40.7%)	7/24 (29.2.%)	2/10 (20.0%)	0.104
Score 11–15		15/27 (55.6%)	14/24 (58.3%)	4/10 (40.0%)	
Score > 15		1/27 (3.7%)	3/24 (12.5%)	4/10 (40.0%)	
Bronchopulmonary Dysplasia at 36 0/7 Weeks	38	11/17 (64.7%)	17/17 (100%)	4/4 (100%)	0.0016
Mild		8/17 (47.1%)	6/17 (35.3%)	0/4 (0%)	
Moderate		1/17 (5.9%)	1/17 (5.9%)	0/4 (0%)	
Severe		2/17 (11.8%)	10 (58.8%)	4/4 (100%)	
On Oxygen Support at 28 Days of Life	40	11/17 (64.7%)	18/18 (100%)	5/5 (100%)	0.0079
Discharged Home on Oxygen Support	37	1/17 (5.9%)	7/16 (43.8%)	4/4 (100%)	0.0004
Median Days Receiving Oxygen at >21% Inspired Air	37	42.0	79.5	130.5	0.001
Median Days Receiving Mechanical Ventilation	37	8.0	34.5	52.0	0.030
Retinopathy of Prematurity	39	8/17 (47.1%)	15/17 (88.2%)	5/5 (100%)	0.0136 0.0087
Stage 1		3/17 (17.6%)	8/17 (47.1%)	3/4 (60.0%)	
Stage 2		1/17 (5.9%)	5/17 (29.4%)	0 (0%)	
Stage 3		4/17 (23.5%)	2/17 (11.8%)	1 (20.0%)	
Stage 4		0/17 (0%)	0 (0%)	1 (20%)	
Necrotizing Enterocolitis	61	1/27 (3.7%)	4/24 (16.7%)	0/10 (0%)	0.181
Spontaneous Intestinal Perforation	61	1/27 (3.7%)	2/24 (8.3%)	0/10 (0%)	0.766
Infant Death	61	10/27 (37.0%)	8/24 (33.3%)	6/10 (60.0%)	0.349

**Table 4 nutrients-16-04041-t004:** Adjusted * odds ratios of clinical outcomes based on timing of enteral feeding initiation at ≥12 vs. <12 h of life for surviving infants (*n* = 37).

Variable	Odds Ratio	95% Confidence Interval		*p*-Value
Bronchopulmonary Dysplasia at 36 0/7 Weeks	12.00	0.62	232.98	0.101
On Oxygen Support at 28 Days of Life	12.00	0.62	232.98	0.101
Bronchopulmonary Dysplasia (Moderate/Severe)	6.99	1.28	38.28	0.025
Discharged Home on Oxygen Support	9.08	1.03	79.81	0.047
Retinopathy of Prematurity	3.72	0.63	22.03	0.147
Retinopathy of Prematurity (≥Stage 2)	0.61	0.11	3.53	0.582

* Logistic regression after adjusting for CRIB II scores as a continuous variable. The analyses model the probability that initiation of enteral feeding is ≥12 h compared to <12 h.

## Data Availability

The raw data supporting the conclusions of this article will be made available by the authors on request.

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
