# Peer review of "Hour of Life at Enteral Feeding Initiation and Associated Clinical Morbidity in Extremely Low-Birth-Weight Infants"

_nutrients, 2024, doi:10.3390/nu16234041_

Round 1

Reviewer 1 Report

Comments and Suggestions for Authors

I have read this paper with great interest, and with a background on perinatal clinical research. I assess the paper as rather confirming or additional evidence, but do have some concerns. I have provided these chronologically. 

abstract: 'promote' suggests causality language, while the level of evidence for this trial is rather association based. 

abstract and full paper: your conclusions or interpretation is rather 'strong' when we reflect on the power, and the different outcome parameters reported. Overall, explorative design, so that I would recommend to soften these conclusions. 

line 37-39: prenatal steroids are neither anymore modifiable at birth. 

Line 46: In example ? 

results, lines 201-203: please check (likely to be removed ?)

line 249-251: please check these sentences, I assume that there is a need for additional editing. 

Author Response

Thank you for your review of this manuscript and for your valuable feedback.  We appreciate the time taken to do so and the input that has helped us enhance the content and clarity of the manuscript.  Our responses are noted below (after a check box) based on your feedback (in bold) and edits made are evident via the "Track Changes" option in Word.

Reviewer: abstract: 'promote' suggests causality language, while the level of evidence for this trial is rather association based.
    This has been modified in the manuscript.

Reviewer: abstract and full paper: your conclusions or interpretation is rather 'strong' when we reflect on the power, and the different outcome parameters reported. Overall, explorative design, so that I would recommend to soften these conclusions. 
    We have modified the verbiage within the abstract and manuscript to soften results, as requested.

Reviewer: line 37-39: prenatal steroids are neither anymore modifiable at birth. 
    Thank you, this has been included in the manuscript.

Reviewer: Line 46: In example ? 
    The language in this section has been modified for clarity.

Reviewer: results, lines 201-203: please check (likely to be removed ?)
    Thank you!  We have removed these lines.

Reviewer: line 249-251: please check these sentences, I assume that there is a need for additional editing.
    This section has been edited in the manuscript.

Reviewer 2 Report

Comments and Suggestions for Authors

The task of the presented study is interesting and important, the manuscript is well written and clearly understandable

Introduction: As several studies and reviews underline the early introduction of enteral feeding even in the smallest premature infants, the approach to the exact timing in hours should be pointed out more intensively. In Europe for instance, it is widely accepted to start enteral feeding in the first 6 hours of life, including immediately buccal colostrum application in this population.

Methods: The design of a retrospective study in combination with a small study cohort is limiting profoundly the level of evidence. On the other hand a mono-centric study may reduce  confounders due to clinical policy. The inclusion of all patients born < 1000g in a period of approximately 4 years seems to reflect the practical approach of a single center. The authors should provide the number of excluded patients also reporting the number drop outs intended to be studied.

Results : The demographics of the study cohort presents an overweight of male sex, what may contribute to an elevated number of morbidities.

Compared to the data of the Vermont-Oxford Network or the German Neonatal Network the mortality of the presented cohort is quite high, even as some morbidities like BPD and NEC. The average days of mechanical ventilation is also higher than in the mentioned networks above.

This may lead to the suspicion, that some main confounders like delivery room management may have mainly influenced the general outcome in the study cohort. Given that the non-significant trends relating to the influence of starting enteral feeding on some outcome criteria are not explained by random, this may partially correct the very low numbers of patients included in the study. Under those circumstances the significant correlation between starting oral feeding < 12 hours of life and pulmomary outcome is an important finding.

Discussion: I would encourage the authors to comment on the current literature comparing their results of early introduction of enteral feeding on BPD and gastrointestinal morbidities, namely the RCT from Salas (Am J Clin Nutr2018 Mar 1;107(3):365-370) and the Review from Miller, (Nutrients 2018 May 31;10(6):707)

Limitations: This very important chapter discussing the limitations of the study meets most aspects effecting the limited level of evidence of the study.

The conclusions drawn form the results of the study are clearly presented.

Overall Comment: Though reporting a retrospective, mono-centric study with a small number of included extremely low birth weight infants, the manuscript provides a hypothesis generating approach for a randomly controlled prospective trial.

Author Response

Thank you for your review of this manuscript and for your valuable feedback.  We appreciate the time taken to do so and the input that has helped us enhance the content and clarity of the manuscript.  Our responses are noted below (after a check box) based on your feedback (in bold) and edits made are evident via the "Track Changes" option in Word.

Reviewer: Introduction: As several studies and reviews underline the early introduction of enteral feeding even in the smallest premature infants, the approach to the exact timing in hours should be pointed out more intensively. In Europe for instance, it is widely accepted to start enteral feeding in the first 6 hours of life, including immediately buccal colostrum application in this population.

    We appreciate this feedback.  We kindly would like to point out that we have included varying research on this in the Discussion section, including one article (reference 34) that may wait up to 10-14 days to provide enteral feeding.  However, we have included more reasons for delayed initiation in the Introduction section.

Reviewer: Methods: The design of a retrospective study in combination with a small study cohort is limiting profoundly the level of evidence. On the other hand a mono-centric study may reduce  confounders due to clinical policy. The inclusion of all patients born < 1000g in a period of approximately 4 years seems to reflect the practical approach of a single center. The authors should provide the number of excluded patients also reporting the number drop outs intended to be studied.

    Thank you for these considerations.  We have noted a smaller sample size within a single center within our Limitations section.  Likewise, we have already included our Methods of inclusion/exclusion and have likewise included the number of surviving vs. non-surviving infants.

Reviewer: Results : The demographics of the study cohort presents an overweight of male sex, what may contribute to an elevated number of morbidities.

    This is an excellent point.  We have included this within the Limitations section.

Reviewer: Compared to the data of the Vermont-Oxford Network or the German Neonatal Network the mortality of the presented cohort is quite high, even as some morbidities like BPD and NEC. The average days of mechanical ventilation is also higher than in the mentioned networks above.

    Agreed—we have already noted this within our Discussion section to comment on the unusually high rate of mortality.

Reviewer: This may lead to the suspicion, that some main confounders like delivery room management may have mainly influenced the general outcome in the study cohort. Given that the non-significant trends relating to the influence of starting enteral feeding on some outcome criteria are not explained by random, this may partially correct the very low numbers of patients included in the study. Under those circumstances the significant correlation between starting oral feeding < 12 hours of life and pulmomary outcome is an important finding.

    Thank you for pointing out this perspective.  We already acknowledge within he Limitations section, that many variables have not been adjusted for.  However, we have included “clinical care within the first hour of life” as an additional noted factor in the Limitations section.

Reviewer: Discussion: I would encourage the authors to comment on the current literature comparing their results of early introduction of enteral feeding on BPD and gastrointestinal morbidities, namely the RCT from Salas (Am J Clin Nutr. 2018 Mar 1;107(3):365-370) and the Review from Miller, (Nutrients 2018 May 31;10(6):707)

   Thank you for these article references.  In reviewing this, we believe our current list of references is appropriate.  However, we have added comments regarding concepts of these articles to the Limitations section.

Reviewer: Limitations: This very important chapter discussing the limitations of the study meets most aspects effecting the limited level of evidence of the study.  The conclusions drawn form the results of the study are clearly presented.

    Thank you for this valuable feedback.

Reviewer 3 Report

Comments and Suggestions for Authors

This is an important and interesting study that will be of significant interest for neonatal practitioners. ELBW infants are a source of worry for neonatal staff , so the findings of this paper will be helpful in all attempts to try and improve outcomes for this vulnerable population of infants. 

My comments are minor:

- In the Introduction, third paragraph, you discuss human milk benefits during enteral feeding. I think many practitioners also use non-nutritive sucking. Please just add this as it is frequently used in the UK . To support this I would cite: : Harding, C., Cockerill, H., Cane, C. & Law, J. (2018). Using non-nutritive sucking to support feeding development for premature infants: A commentary on approaches and current practice. Journal of Pediatric Rehabilitation Medicine, 11(3), pp. 147-152. doi: 10.3233/prm-170442.

This paper clearly discusses non-nutritive sucking pre introducing oral feeding, i.e. when infants are receiving tube feeds.

- Please use "Data were" instead of "Data was".

- Could you clarify if all infant participants had NGTs, or if any had OGTs? If there were any difference, please add to relevant tables. 

- In your discussion, can you please consider the potential and longer term benefits in relation to the progression of oral feeding? It may be , of course that you don't know as yet, but it would be useful to consider this. 

Author Response

Thank you for your review of this manuscript and for your valuable feedback.  We appreciate the time taken to do so and the input that has helped us enhance the content and clarity of the manuscript.  Our responses are noted below (after a check box) based on your feedback (in bold) and edits made are evident via the "Track Changes" option in Word.

Reviewer:  In the Introduction, third paragraph, you discuss human milk benefits during enteral feeding. I think many practitioners also use non-nutritive sucking. Please just add this as it is frequently used in the UK . To support this I would cite: : Harding, C., Cockerill, H., Cane, C. & Law, J. (2018). Using non-nutritive sucking to support feeding development for premature infants: A commentary on approaches and current practice. Journal of Pediatric Rehabilitation Medicine, 11(3), pp. 147-152. doi: 10.3233/prm-170442.

    Thank you for this article reference.  We prefer reference #13, which is more aligned with our unit practices to provide oral cares using human milk.  However, we have noted this intervention more prominently in the Discussion and Limitations section.

Reviewer: This paper clearly discusses non-nutritive sucking pre introducing oral feeding, i.e. when infants are receiving tube feeds.

    We provide these ELBW infants with oral cares early in life and not necessarily “non-nutritive sucking”.

Reviewer: Please use "Data were" instead of "Data was".
    These changes have been made within the manuscript.

Reviewer: Could you clarify if all infant participants had NGTs, or if any had OGTs? If there were any difference, please add to relevant tables.

    This is a great inquiry.  The use of an NGT vs. OGT was at the determination of the placing provider, depending on type of respiratory support used, nare size, etc.  This was also likely to change over time based on clinical status and the variables listed above, which makes it difficult to categorize.

Reviewer: your discussion, can you please consider the potential and longer term benefits in relation to the progression of oral feeding? It may be , of course that you don't know as yet, but it would be useful to consider this.

    Thank you, the Discussion section of the manuscript has been modified to include this as we don’t currently know what the relationship is.